# Aqueous Binary Mixtures of Stearic Acid and Its Hydroxylated Counterpart 12-Hydroxystearic Acid: Cascade of Morphological Transitions at Room Temperature

**DOI:** 10.3390/molecules28114336

**Published:** 2023-05-25

**Authors:** Maëva Almeida, Daniel Dudzinski, Catherine Amiel, Jean-Michel Guigner, Sylvain Prévost, Clémence Le Coeur, Fabrice Cousin

**Affiliations:** 1Institut Chimie et Materiaux Paris Est, Université Paris Est Créteil, CNRS, UMR 7182, 2 Rue Henri Dunant, 94320 Thiais, France; maeva.ferreira-almeida@u-pec.fr (M.A.); catherine.amiel-guenoun@cnrs.fr (C.A.); 2Laboratoire Léon Brillouin, Université Paris-Saclay, CEA-CNRS UMR 12 CEA Saclay, 91191 Gif sur Yvette, France; daniel.dudzinski@cea.fr; 3Institut de Minéralogie, de Physique des Matériaux et de Cosmochimie (IMPMC)-IRD-MNHN, Sorbonne Université & CNRS, UMR 7590, CEDEX 05, 75252 Paris, France; jean-michel.guigner@sorbonne-universite.fr; 4Institut Laue-Langevin—71 Avenue des Martyrs, CS 20156, CEDEX 9, 38042 Grenoble, France; prevost@ill.fr

**Keywords:** hydroxystearic acid, stearic acid, self-assembly, mixtures, small angle neutron scattering

## Abstract

Here, we describe the behavior of mixtures of stearic acid (SA) and its hydroxylated counterpart 12-hydroxystearic acid (12-HSA) in aqueous mixtures at room temperature as a function of the 12-HSA/SA mole ratio R. The morphologies of the self-assembled aggregates are obtained through a multi-structural approach that combines confocal and cryo-TEM microscopies with small-angle neutron scattering (SANS) and wide-angle X-ray scattering (WAXS) measurements, coupled with rheology measurements. Fatty acids are solubilized by an excess of ethanolamine counterions, so that their heads are negatively charged. A clear trend towards partitioning between the two types of fatty acids is observed, presumably driven by the favorable formation of a H-bond network between hydroxyl OH function on the 12th carbon. For all R, the self-assembled structures are locally lamellar, with bilayers composed of crystallized and strongly interdigitated fatty acids. At high R, multilamellar tubes are formed. The doping via a low amount of SA molecules slightly modifies the dimensions of the tubes and decreases the bilayer rigidity. The solutions have a gel-like behavior. At intermediate R, tubes coexist in solution with helical ribbons. At low R, local partitioning also occurs, and the architecture of the self-assemblies associates the two morphologies of the pure fatty acids systems: they are faceted objects with planar domains enriched in SA molecules, capped with curved domains enriched in 12-HSA molecules. The rigidity of the bilayers is strongly increased, as well their storage modulus. The solutions remain, however, viscous fluids in this regime.

## 1. Introduction

The climatic situation of the planet and the depletion of fossil resources has led to an increase in the need for the use of environmentally friendly molecules. In such a context, fatty acids, i.e., carboxylic acids with an aliphatic chain, are an alternative to oil-based surfactants, such as green surfactants, since they are mostly extracted from plants. Indeed, they are probably the oldest surfactants used by humans, as their use as soaps has been reported since ancient times and have been increasingly used by the industry [1]. The dispersion in aqueous solution of the fatty acids with long chains (C_14_ to C_22_) remains, however, a challenge, since they have a high Krafft temperature with conventional metallic alkali counterions [2]. This has recently driven a huge effort to achieve such a dispersion, owing to both (i) the large availability of the biomass of C14 tetradecanoic acid, C16 hexadecanoic acid C18 and octadecanoic acid, better known by their respective common names, namely myristic acid, palmitic acid and stearic acid (SA, see scheme Figure 1B), and (ii) to the various morphologies they may form once dispersed in water that provide them with a huge potential for their use as basic blocks for the design of stimuli-responsive systems, such as foams or emulsions [1,3].

Aside from the possibility of a chemical modification from the fairly reactive carboxylate group, the main strategy that enables the dispersion in water relies on the addition of a species that is prone to interacting with the carboxylate head, according to its ionization state, either by hydrogen-bonding when the head is in its COOH form or by electrostatics and ion-pairing interactions when the head is in its COO^−^ form. There are two main methods that have been reported, depending on whether this added species is a surfactant itself or not.

The first method is the use of cationic surfactants, which usually contain an amino head that allows the formation catanionic self-assemblies, i.e., a mixture of cationic and anionic surfactants, upon mixing with fatty acids. Their dispersion originates from the combination of both electrostatic interactions between the fatty acid and the oppositely charged cationic surfactant and hydrogen bonding that give rise to the formation of catanionic surfactant pairs with amphiphilic properties, as explained in the review from Fameau and Zemb [3]. By playing on parameters such as the type of polar group or the ratio between surfactants, it is possible to design aggregate structures with a very broad morphology (mixed micelles, lamellar phases, vesicles, bicontinuous structure) [4,5,6], the most spectacular being obtained via the myristic/cetyltrimethylammonium hydroxide system in which flat nanodiscs of finite size [7], regular hollow icosahedra [8] or very robust vesicles that resist dialysis [9] have been reported.

The second method is ion pairing with, e.g., an alkyl-amine that decreases the Krafft temperature, which, for instance, is obtained through the simple neutralization of the unsaturated fatty acids from C14 to C22 via tetrabutylammonium hydroxide, leading to small micelles [10]. Such a strategy has been successfully tested for a large range of amino counterion/fatty acid systems [11,12,13,14,15,16,17,18,19,20,21,22]. It is worth mentioning that solubilization at low temperature can also be achieved via chaotropic counterions such as K^+^ or Cs^+^ [4,10,12,15]. The morphologies of the self-assembled aggregates mainly depend on three parameters: (i) the size and hydrophobicity of the counterion; (ii) the state of crystallinity of the alkyl chains and, therefore, the temperature; and (iii) the degree of ionization of the carboxylated head that depends on pH [23]. Indeed, the self-assemblies that are formed when fatty acids are in their disordered liquid crystal state are more flexible than when they are on their Lβ-ordered crystalline phase, which increases their curvature. The melting transition between the two states may thus induce a morphological transition. Interactions between neighboring heads are strongly dependent on their ionization state because there is competition between H-bonding that promotes their approach, which is prompted to occur when they are on their protonated COOH form, and electrostatic repulsions when they are on their COO^−^ form, which moves them apart. The overall resulting subtle interplay of interactions tunes the packing parameters of the fatty acids and, therefore, their morphology. The counterion/fatty acid ratio is then a key parameter, since it fixes the pH. Lamellar self-assemblies are obtained when pH is close to the pKa of fatty acids, with a large polymorphism (faceted objects, planar lamellar phases and unilamellar or mutilamellar vesicles [14,15,18,19]) and spherical micelles at pH > pKa [14,15,18,19,20]. However, it has been reported that the use of a chaotropic ion (Cs^+^ or K^+^) makes it possible to extend the lamellar domain towards large pH thanks to a strong screening of the COO^−^ heads [15]. Additionally, the fine-tuning of the interactions makes it possible to obtain the packing parameter, thus allowing the formation of vermiform micelles [16].

The introduction of a hydroxyl function OH onto the alkyl chains provides a huge potential for fatty acids compared to their saturated non-modified counterpart because it results in three functionalities in the molecules: chirality, a second hydrophilic center and the possibility of making H bonds between the secondary OH groups. At the air/water interface, hydroxylated fatty acids, which are sometimes called “bihydrophilic”, indeed show a specific behavior when the OH group is located sufficiently far from the head group (typically at the seventh carbon of the alkyl chain or further) [24,25]. At intermediate surface pressure, in between the condensed liquid phase and the gas phase, they adopt a specific conformation where they adsorb to the interface with their two hydrophilic centers in contact with water, which gives rise to a very large plateau in the surface pressure versus surface area isotherm [24,25,26]. At a large surface pressure, the molecules are not elongated in a straight manner but may instead present a kink at their second OH group. Moreover, in the condensed phase, the formation of a hydrogen bond network [27] makes the monolayer very elastic since their surface viscosity can reach values that are three orders of magnitude larger than those of the gas phase [26].

Among these hydroxylated fatty acids, 12-hydroxy stearic acid (12-HSA, see scheme Figure 1A) is widely used because it is easily obtained from the hydrogenation of the double bond of rinoleic acid, which makes up 90% of castor oil extracted from castor beans [2]. Owing to its possibility of making H bonds in an apolar medium, it is, in particular, very widely used as an organogelator, either in its pure form or in a chemically modified one [2,28,29,30,31,32,33,34,35]. In aqueous solution, the chirality of 12-HSA molecules enables them to create remarkable self-assemblies in the presence of alkoxyamines counterions or their derivatives: twisted ribbons [36] or multilamellar tubes [37]. These multilamellar tubes have a length of about 10 µm, a diameter of 0.6 µm and their core is composed of a few (~3–6) stacked bilayers separated by water and can be obtained from a large set of counterions [38,39]. Their formation likely comes from the rolling of helical ribbons. While there is a large body of literature on the formation of single wall tubes based on organic species such as peptides [40], the architecture of multilamellar tubes is, by contrast, sparsely discussed and has only been garnered attention recently, along with complexes of ionic surfactants and cyclodextrins [41,42,43,44,45].

A unique specificity of the multilamellar tubes of 12-HSA is their ability to swell and de-swell over a wide range of temperatures, resulting in a tunable diameter [46]. At high temperatures, they melt into small micelles above a threshold melting temperature [38] that depends on the counterion/12-HSA ratio [39]: in excess of counterions, the transition is mostly driven by the melting of the chains, whereas at lower contents of counterions, where H-bonding occurs between non-dissociated heads, it is driven by both the chain-melting process and the surface-melting process. The formation of the tubes is a robust process as they can be obtained under a wide range of physico-chemical conditions (concentrations, molar ratio, pH, ionic strength, addition of ethanol or doping by fatty acids) [11]. Aqueous suspensions have an hydrogel behavior induced by entanglements between tubes [2], since their viscosity is larger by three to five orders of magnitude, depending on swelling, than that of the regime of small micelles at high temperatures [39,47]. This opens the way for the design of thermo-responsive systems. For instance, such a large change of viscosity, associated with the strong elasticity of the 12-HSA monolayers, has enabled the design of ultra-stable and responsive aqueous foams [48], with a tunable temperature for destabilization [49]. The loading of tubes via molecules of biomedical interest for drug delivery applications has been demonstrated [50].

We propose here a new strategy to design the self-assemblies of long-chain fatty acids in aqueous solutions with a tunable structures and a strong potential for the design of thermo-responsive systems. It is based on the formation of binary mixture of surfactants that have the same polar head, namely a saturated long-chain fatty acid and its hydroxylated counterpart. The behavior will then be driven by specific interactions between both surfactants that occur at the level of the alkyl chains and not through the interactions between the polar heads, as occurs in the usual surfactant mixtures, such as catanionic systems. To this end, we choose to work on the stearic acid/12-hydroxystearic acid pair, as both fatty acids have been solubilized in past works under the same conditions with ethanolamine counterion, in which they formed very different self-assemblies at low temperatures: multilamellar tubes for 12-HAS [38] and planar lamella and vesicles for the SA/ethanolamine systems [14]. Moreover, both systems melt into micelles at high, but different, temperatures [14,39]. We also built on the seminal work of Fameau et al. [11], which showed that 12-HSA molecules still form tubes at room temperature upon the introduction of a small amount of SA but turn into ribbons at larger SA contents, a behavior that was hypothesized as resulting from a local partitioning between the two kinds of fatty acids. The structure of the binary self-assemblies was, however, poorly described, as it was only obtained through phase-contrast microscopy. We will, in particular, probe the regime of SA doped by a few 12-HSA molecules, which has never been studied in solution to the best of our knowledge. Synergistic effects may be anticipated since the 2D phase diagram of the monolayers of mixtures SA/12-HSA has a eutectic point [51].

We focus on the behavior of the system at room temperature, exhaustively probing its structural and rheological properties over the full range of R (12-HSA/SA mole ratio). The multi-structural approach combines confocal microscopy, SANS, WAXS and CryoTEM measurements and is coupled with rheology measurements.

## 2. Results

### 2.1. Macroscopic Aspect

In order to obtain the mixtures, we chose to use a large excess of ethanolamine counterions so that all fatty acid heads would be ionized. This prompted us to work at *r* = 0.2, where $r=n_{fatty\;acid}/(n_{fatty\;acid}+n_{ethanolamine)}$, where the tubes/micelles transition for pure 12-HSA solution is ~40 °C [3,39]. The overall concentration of fatty acids was fixed at 2 wt% of surfactants in water, as in references [3,11,38,39] on pure solutions of 12-HSA, so the mixtures can be compared to these. The stock-solutions of 12-HSA and SA were prepared at 70 °C, following a protocol previously developed for pure 12-HSA solutions (see the Section 4). Both 12-HSA and SA solutions were clear at such a high temperatures, suggesting that the formation of micelles was in agreement with previous articles on SA [14] and 12-HAS [38,39]; they were mixed at 70 °C, a temperature at which the resulting mixtures remained homogeneous and clear. The whole range of ratio R, where R = n_HSA_/n_SA_ + n_HSA_, was examined, the R = 0 sample being the pure SA and the R = 1 sample being the pure 12-HSA. For all samples, the pH of the mixtures were constant (pH = 10.85 ± 0.15, see Table 1) at a value much larger than the pKa of the carboxylate groups (pKa of pure 12-hydroxy stearic acid is 4.75 and around 8 in presence of excess alkanolamines [2]).

When cooling down the mixtures to an ambient temperature, all samples turned a turbid white, revealing the formation of lamellar rather than vesicles (Figure 2). At R = 0.9 and R = 1, the turbidity of samples was more intense, suggesting the formation of larger objects than for other ratios. All the samples looked homogeneous to the naked eye with the exception of R = 0.4, which revealed some heterogeneities at macroscopic scale with respect to turbidity (see inset of Figure 2 for R = 0.4). The sample R = 0.5 also shows visual heterogeneities, but these are not easily captured through photography. All the samples have an apparently high viscosity.

### 2.2. Confocal Microscopy

Confocal microscopy of samples has been performed on the whole range of ratio R using a hydrophobic dye, Nile Red. The Nile Red concentration is negligible with respect to the fatty acid concentration (see Section 4), so that we assume that the morphology of the self-assembled structures is not modified by its introduction. Figure 3 shows some representative images of samples for all R ratios examined using ×100 magnification. On the images, Nile Red is concentrated in the black areas that do correspond to the hydrophobic regions of the self-assembled structures made of the alkyl chains of the fatty acids, whereas the white domains do not contain the fluorophores.

Figure 3I shows the image obtained for R = 1 (pure 12-has), which reveals the presence of disordered thin and rigid rods with a length of around 20 μm. These rods correspond to the multilamellar tubes resulting from the self-assembly of 12-HSA molecules [39]. The organization of these tubes appears disordered, without the formation of any nematic order. When R decreases from 1 to 0.75 upon the introduction of a small amount of SA (Figure 3H,G), and the self-assembled structures conserve their rodlike shapes. It appears, however, that the overall length of the rods decreases progressively, with a distribution of lengths that become more polydisperse when R decreases. We did not perform a statistical study of these variations in tube length in this study of large R ratios because the images are transversal cross-sections of a 3D sample.

At R = 0.75, it also appears that a very small number of curved objects coexist with the long tubes. This trend is confirmed when decreasing R in the regime of intermediate R (R = 0.6, R = 0.5). Indeed, for R = 0.6 (Figure 3F), the rigid thin polydisperse rods now coexist with an important number of objects with a large curvature. For R = 0.5, during stoichiometry, fatty acids self-assemble into long, bent tubes that are able to reach a much longer length that the 20 μm obtained at R = 1 and extend to up to 60 μm (Figure 3E). These bent tubes appear to be made up of two types of domains, with rigid parts bound to each other by curved parts. In this regime of intermediate R, the structure of fatty acid self-assemblies thus evolves continuously from bent tubes to straight ones through R. In their seminal study, where they introduced a significant amount of SA into aqueous solutions of 12-HSA, Fameau et al. [11] observed, through phase contrast microscopy, the formation of twisted ribbons that are the precursors of multilamellar tubes. Confocal microscopy does not allow the discrimination between multilamellar tubes and ribbons; however, we hypothesize that the rigid parts correspond to the tubes and the bent ones to ribbons.

At low R, where SA is in excess, the structure is completely different and images are less simple to interpret (see Figure 3B–D, which corresponds, respectively, to R = 0.4, R = 0.3 and R = 0.2). There are no longer thin black lines in which all the hydrophobic Nile Red molecules concentrate, as was the case for higher R, but instead, 2D domains of variable levels of grey are extant. Most of these domains have straight sides, suggesting the formation of faceted objects, connected to each other by either edges or by curved parts. Some other domains display a more spherical shape. These domains are all made of multilamellar bilayers, as demonstrated by SANS, as we will show in a later section. Thus, they have different distributions of thicknesses and orientations, which give rise to this variation in the grey level from one domain to another. The fact that the majority of their sides are either straight or circular suggests that they are made of planar lamellar phases connected by spherical lamellar phases, which are possibly parts of multilamellar tubes. The images also suggest that these domains are strongly entangled. Indeed, when following the temporal evolution of the system during the experiment, it appeared that these domains do not move individually but move collectively. This collective motion of domains is consistent with the fact that they are either connected to each other and/or strongly entangled.

Finally, for R = 0 (pure SA), there are only small fluctuations in the color of the images, with a typical size of fluctuations of the order of a few microns, although SANS experiments shown later will unambiguously demonstrate that this sample contains planar lamellae. Since all lamellar phase domains contain the same level of fluorophores on average, the only variations of the grey level within the images come from the orientation of the 2D lamellar domains. The image is thus consistent with a structural organization of the planar lamellar phase domains with random orientations that occupy all space, in agreement with the literature [15].

### 2.3. Cryogenic Transmission Electron Microscopy (Cryo-TEM)

The pictures of samples were obtained through cryogenic transmission electron microscopy for pure 12-HSA (R = 1), pure SA (R = 0), the intermediate content of both kinds of molecules (R = 0.4) and the low content of SA (R = 0.1). Samples were left at room temperature (around 20 °C) for 24 h to equilibrate before the freezing step. The final thickness of the frozen cut sample is around one micrometer, which is much lower than the length of the tubes of the order of 20 μm, observed by confocal microscopy. This forces them to lie on the grid’s surface.

For R = 1, long straight rectangular objects are visible with a width of ~500 nm and a length that is at least 8 μm, as they exceed the size that corresponds to the image section at the lowest magnification utilized, in agreement with the length of 20 μm determined via confocal microscopy (Figure 4A–C). In all of these rectangles, there is a continuous gradient within the grey scale from the outer of the edges of the rectangle, which are lighter, to the inner areas of the rectangle, which are darker, demonstrating a continuous inward variation in their thickness from the exterior to the interior. Even though the images are 2D, it is demonstrated that these rectangular shapes correspond to cylindrical tubes. There are some parallel dark lines within the tubes, which are parallel to their edges, which is caused by the presence of lamellas. Although SANS experiments demonstrate that the inter-lamellar distance is constant (see later), such a distance between lamellas does not appear to be strictly constant in the images, but this is an effect of projection from the 3D self-assemblies to the 2D images. Moreover, the dark centers of the rectangles tubes do not display any lamella. This is also consistent with the formation of multilamellar tubes, as such a dark center corresponds to the top of the adsorbed cylinders, where lamella are parallel to the surface. Thus, the exact estimate of the number of lamella is not possible from these cryo-TEM images, yet they reveal that there are at least five or six lamella per tube. Since the tubes lie on the surface, there are no images of the cross-sections of the tubes, but they would resemble onions. Similar pictures have been obtained via cryo-TEM on the multilamellar tubes of SDS@2β-CD [42]. In summary, these cryo-TEM images are fully consistent with the structure proposed in the literature for 12-HSA self-assemblies as long rigid multilamellar tubes. They also show that the radius of these tubes are rather monodispersed, since their widths are fairly constant from one tube to another. However, a more interesting feature concerns their ends, which are open and not capped, as it is unambiguously visible in some pictures (Figure 4B). This had not previously been established in the literature.

For pure SA (R = 0), while turbidity and SANS (see later) unambiguously demonstrate that large self-assembled lamellar aggregates do exist in the solution, they are not visible at first sight on the grid. There are areas of various size, which are more or less dark, separated by straight edges that presumably correspond to steps between domains of different thicknesses (Figure 4L,M). Such areas derive from the covering of the surface by surfactants molecules, as proven by the presence of cracks, which are depleted in molecules, since their light grey level corresponds to the pure grid, as shown in Figure 4N. It is likely that planar lamellar phases with a variable number of lamellae, but probably a low number, are deposited on the surface and lay parallel to it. The images thus depict a top-down view of such lamella and the steps correspond to a change in lamella number from one domain to another. The darker the area, the larger the number of lamella. Similar pictures have also been obtained via cryo-TEM on the Lα lamella phase on fatty acids with an excess of alkali [15].

For R = 0.4, there are also long rigid multilamellar tubes with open ends, as for the R = 1 case with the same order of magnitude of the radius (Figure 4D–F) and a length that still exceeds 10 microns (Figure 4G). However, compared to the R = 1 case, the number of lamella seems to be slightly reduced on average and varies more significantly from one tube to another. It also appears that parts of the tubes are slightly curved. This is in line with what was observed via confocal microscopy at R = 0.5 with long 1D objects with rigid and soft domains. This suggests that the parts of the objects analyzed via cryo-TEM are twisted ribbons and not multilamellar tubes, although the cryo-TEM images do not allow the arbitration between both structures as they provide the same image due to projection. The curved parts would result from the ribbons with the lowest rigidity.

For R = 0.1, there are no longer multilamellar tubes, but the images still reveal the existence of lamellar structures with different morphologies and a varying number of lamella. The most striking feature concerning the curvature of these lamella is that they appear either straight, which may account for planar lamella or pieces of large tubes, or spherical, suggesting truncated onion vesicles (Figure 4H,J). Such straight and spherical lamella eventually merge to form a single multilamellar objects (Figure 4I). Regarding the number of lamella, it ranges from a single lamella up to multiple with the same order of magnitudes as the tubes at R = 1. Additionally, images depict the dark areas and the steps, revealing the presence of planar lamellas sitting on the surface, as seen in the R = 0 case (Figure 4K). The samples are thus composed of different forms of self-assemblies that partially fuse with different curvatures. It is thus likely that these different kinds of self-assembled aggregates do not all contain the same amounts of SA and 12-HSA molecules, i.e., a partitioning between the two surfactants exists at the local scale.

### 2.4. Structure at Local Scale via SANS

The structure of self-assembled surfactants was determined via SANS for the whole range of R (Figure 5A). SANS examines a range of distances from ca. 10 Å to 1 µm in the solution, with no artifacts from deposition on a 2D surface and no addition of an extra probe. It does require to work in heavy water D_2_O to create a contrast with the hydrogen-rich molecules and decrease the large incoherent scattering of ^1^H, which acts as a constant background for SANS. Data are presented as intensity versus q, the magnitude of the wavevector, which is expressed at a reciprocal length scale. In an initial approximation, intensity features at a given q-value can be associated with the characteristic dimensions *d* based on the Bragg relationship *d*~2π/q.

The SANS scattering spectra of the pure SA sample (R = 0) display all the characteristic features of a 2D lamellar phase: (*i*) a q^−2^ scattering decay in the low q region, typical of the scattering of a 2D object and accounts for the planar shape of lamella; (*ii*) a strong correlation peak at q_0_ = 0.0109 Å^−1^, followed by its harmonics at n·q_0_ (*n* = 2, 3, 4…), which accounts for the Bragg peak associated with the interlamellar distance *d*_inter_, whose order of magnitude is ~575 Å (2π/q_0_); and (*iii*) a so-called form factor oscillation at around 0.25 Å^−1^, originating from the cross-section of the lamellae and enabling the determination of its thickness at ~25 Å (2π/0.25). Such scattering spectra unambiguously demonstrate that the sample is made of a lamellar phase, in agreement with previous results [14], which deduced the formation of a lamellar phase through birefringence measurements, even though they were not visible on the cryo-TEM pictures.

The pure 12-HSA system (R = 1) also displays the characteristic features of lamellar phases at the intermediate and large q, (i.e., a correlation peak followed by its harmonics, oscillation at large q). It was immediately obvious, however, that interlamellar thickness is strongly reduced compared to that in the pure SA case and interactions between lamella are different from those in the pure SA case, since q_0_R = 1_ is strongly shifted towards low q with respect to q_0_R = 0_ at ~0.025 Å^−1^ (*d*_inter_ = 240 Å). The correlation peak is narrower, its harmonics are visible up to the sixth order, indicating a more regular periodicity, i.e., bilayers fluctuate less in the R = 1 case and/or comprise more stacked bilayers in the tube wall than those forming the SA lamellar phase. The thickness of the lamella is similar to that of the pure SA case, since the form factor oscillation is also at 0.25 Å^−1^. The main difference to the SA case appears in the low q region, where the overall scattering decays as q^−3^ and no longer as q^−2^ and shows a well-defined marked oscillation at 1.2 10^−3^ Å^−1^, resulting from the form factor of the tube diameter. The highlighting of such an oscillation was made possible by the extended low q-range of the D11 diffractometer and was not yet reported on the scattering curves of the same system in the literature as they were all limited to a reduced q-range. The overall q^−3^ decay has, however, already been obtained on the scattering of hollow tubes and multilamellar tubes on other systems [42,43,45,52]. Assuming that tubes are rigid, such q^−3^ can be explained by a decoupling approximation from a tube in its length (∝ 1/q) and by its 2D discotic section (∝ 1/q^2^). In the case of an imogolite hollow tube, Paineau et al. [52], proposed fitting the scattering form factor through the equation $I{\left(q\right)}_{tube}\propto \frac{1}{q}[{\left(\Delta \rho \left(F_{R_{ext}}-F_{R_{int}}\right)\right]}^{2}$, where Δ*ρ* is the difference in scattering density between the tubes and the solvent; R_ext_ and R_int_ are, respectively, the external and internal diameters of the hollow tubes; and F is the Fourier transform of the projection or a full cylinder of radius R along its axis $F_{R}\left(q\right)\propto \frac{RJ_{1}\left(QR\right)}{Q}$, with J_1_ being the first-order cylindrical Bessel function. The extension to orientationally averaged multilamellar tubes was proposed in Reference [42] for the fitting of the scattering of multilamellar tubes made of SDS@2β-CD complexes. It is difficult to fit by such a model here because the number of lamella is unknown. We therefore fitted the low q part of the scattering spectra with a model of hollow tube with a single bilayer of fatty acids, which allows the fitting of the minima and the recovery of the q^−3^ decay (Figure 5A). The tube radius R_tube_ was estimated at 200 nm. Such a model, however, overestimates the intensity in the very low q part of the spectra with respect to the experimental data. This is due to the high scattering intensity in this very low q region at ~10^6^–10^7^ cm^−1^, resulting in multiple scattering events that, in practice, lower the experimental measured pattern from its true value.

For the other R ratio examined, the SANS scattering demonstrate the formation of lamellar phases in every case, yet several systems can be clearly evidenced.

In the instance of the low content of SA molecules, in which rigid tubes were observed through confocal microscopy (1 > R ≥ 0.75), the scattering spectra are very similar to those of the pure 12-HSA system at R = 1. The oscillation arising from the form factor of the tube radius and an overall q^−3^ scattering decay at low q, as well as the correlation peak of the interlamellar phase and its harmonics, are present at the same positions and their amplitudes and widths display the same orders of magnitude. The mixture of molecules thus self-associated into multilamellar tubes have a very similar structures, as seen in the pure 12-HSA system, revealing that these tubes are able to accommodate a significant number of SA molecules. These SA molecules insert themselves into the 12-HSA self-assemblies without disturbing the structures of the tubes. The tube radius increases slightly with respect to the R = 1 case, with 210 nm for R = 0.9 and 250 nm for R = 0.75.

At intermediate R, for R = 0.6 and R = 0.5, all of the scattering features of multilamellar tubes are recovered, except for the low q oscillation of the tube form factor that is no longer present, the scattering decays similarly to q^−3^ over the whole low q-range examined. It this thus likely that the distribution of the outer diameters of the tubes becomes very broad, which leads to the disappearance of the oscillation. This likely results from the fact that the self-assemblies are made of both multilamellar tubes and twisted ribbons, and the respective rigid and soft domains of the tubes were observed through confocal and cryo-TEM microscopies, in agreement with Reference [11]. Even if the interlamellar distance is well defined, it is also possible that both rigid and curved domains do not contain the same number of lamella, as this depends on the degree to which the lamella rolled into the ribbons.

At R = 0.4, the behavior is different. Although the scattering features are similar to those of the intermediate R case, the correlation peak and harmonics are strongly attenuated and shifted towards a larger q, which demonstrates that interactions between the lamella within tubes are largely different for such R. However, it should be kept in mind that sample phases undergo a partial phase separation at the macroscopic scale for R and that the two phases are present in the measurement cuvette, since it is filled by an homogenous solution at 70 °C that has been cooled down. Given that the neutron beam illuminates both phases (with a beam cross-section of around 1 cm^−2^), it is difficult to go deeper into SANS analysis in such a sample.

For R ≤ 0.25, with a large content in the SA molecules, the scattering curves markedly differ from the other cases. First, the correlation peak and its harmonics are much narrower than in the other cases, demonstrating either stronger repulsions between lamella, thus leading to a more regular periodicity, or a larger number of lamella per stack. Second, the scattering decays similarly to q^−4^ in the low q regime and no longer similarly to q^−3^, even if cryo-TEM images show that there are still tubes in sample. This is the sample where confocal microscopy shows faceted objects and cryo-TEM straight rigid lamella and spherically curved lamella joined to each other. Neutrons thus interact with the surfaces of very different orientations within the typical scale of the coherence length of neutrons (a few micrometers), which enables the recovery of the q^−4^ Porod’s law. We postulate that in this regime, where SA molecules are in excess, there is a partitioning between the two types of surfactants within the self-assemblies with 2D lamellar phases of SA that are capped by partial pieces of vesicles (unilamellar or multilamellar), made up of 12-HSA molecules.

These first qualitative descriptions pointed out that there is a significant evolution of the broadening of correlation peaks as a function of R. This may occur either because bilayers fluctuate with different amplitudes or because the number of stacked bilayers in the tube walls/lamellar phase changes. In order to obtain a refined quantitative description, all scattering curves in the intermediate and large q-region were fitted using a model proposed by Nallet et al. [53], which considers a form factor of the lamella and a structure factor between lamella; both the number of stacked bilayers N_lam_ and the Caillé parameter (η) were adjusted, accounting for the thermal fluctuations of the bilayers. Such a model allows us to determine the structural parameters of the lamella (thickness, d-spacing *d*_inter_, rigidity from the Caillé parameter). η is essentially determined from the region of the spectrum where the Bragg peaks arise at intermediate q and the thickness of the lamella from the oscillation at large q, which enables us to decouple these parameters fairly confidently in the fitting process. It is, however, difficult to unambiguously decouple N_lam_ from the Caillé parameter since both parameters play on the Bragg peak amplitudes. We thus varied both parameters when minimizing χ^2^ during the fitting procedure, with a range of N_lam_ varying from 3 to 8. This maximal tested N_lam_ was chosen from geometrical constraints, given that N_lam_ multiplied by the interlamellar spacing *d*_inter_ cannot exceed the tube radius. The values of N_lam_, corresponding to the best fits, are shown in Appendix A. For samples with a large R, N_lam_ has a value of 3 or 4, in agreement with cryo-TEM images and previous data reported in the literature of four stacked bilayers for a pure 12-HSA system [37] and reaches a maximal value of 7 at low R.

This model is able to satisfactorily fit all the data of which multilamellar objects were formed at 20 °C for all R ratios (see Figure 5A). The low q part of the curves was not fitted as it displays different behaviors from one sample to another (q^−2^ versus q^−3^ versus q^−4^), which is derived from either planar objects, tubes or ribbons or faceted objects. These different behaviors at low q are highlighted in Appendix A, which show the experimental data with a I(q)q^α^ versus q representation, with α = 2, α = 3 and α = 4, respectively). In the case of tubes that have a monodisperse radius, the radius was obtained from the fit of the low q part via a model of a hollow tube, as explained preivously. The quantitative values obtained for the fits of *d*_inter_, η and the bilayer thickness as a function of R are shown in Appendix A.

The bilayer thickness has an almost constant value of ~24 Å in all cases—the slight differences from one sample to another are within the fit precision range—which is slightly larger than the chain lengths of the fatty acids (21 Å) [38], which are similar regardless of whether 12-HSA or SA molecules are present. Such a low value of bilayer thickness demonstrates that fatty acid chains are strongly interdigitated.

*d*_inter_ is much larger for pure SA (*d*_inter_ = 575 Å at R = 0) than for pure 12-HSA (*d*_inter_ = 240 Å at R = 1) and varies almost continuously with R between these two values with two clear-cut regimes: (*i*) at a large R (>0.5), in the system where multilamellar tubes are formed, this interlamellar distance varies only very slightly from one R to another, (*ii*) while at a lower R, it decreases continuously with R as a linear decrease in the low R system where partitioning between two types of fatty acids occurs (0.1 ≤ R ≤ 0.25). In this last low R system, one may hypothesize that this experimentally obtained *d*_inter_ is simply effective interlamellar spacing, resulting from the linear combination of the pure planar lamellar spacing of large *d*_inter_ and the pieces of multilamellar tubes of low *d*_inter_. Such a coexistence of zones of very different *d*_inter_ is, however, very unlikely because it would have led to the broadening of the peak, and the reverse is observed, up to the appearance of two separate correlation peaks. Moreover, the objects depicted in cryo-TEM and confocal microscopy appear continuously. Finally, *d*_inter_ is much smaller than for the other samples at R = 0.4, but there are uncertainties regarding the measurement of such a sample, as noted previously.

The evolution of η is shown in Figure 5B. The bilayers are as rigid as the Caillé parameter is low. For such a parameter, we refrain from discussing slight differences from one sample to another too deeply, since N_lam_ and η are coupled parameters during fitting, which gives an intrinsic uncertainty regarding the quantitative fitted values of η. To assess whether such quantitative values were strongly influenced by N_lam_, we also fitted all scattering curves of Figure 5B with a fixed number of N_lam_ that was equal to 4. It appears that the values of η obtained in this latter case are always very close to those obtained when N_lam_ is a floating parameter during fitting (see Appendix A). The large variations of η observed in Figure 5B are thus representative of variations in the rigidity of layers. It appears that the partial doping of the 12-HSA multilamellar tubes through SA induces a progressive decrease in the rigidity of the bilayer since η progressively increases from 0.11 at R = 1 to η = 0.18 at R = 0.75 when progressively introducing small amounts of 12-HSA molecules. A constant value of ~0.18 at the intermediate R is thus retained, which is also the rigidity obtained for pure SA lamellar phases at R = 0. Surprisingly, in the low R system (0.1 ≤ R ≤ 0.25), the bilayers are much more rigid, since the value of the Caillé parameter is around five times smaller, with an almost constant value of 0.03.

### 2.5. Wide Angle X-ray Scattering

WAXS measurements were performed to determine whether the fatty acids chains within the bilayers were fluid or in a gelled state for the whole range of 12-HSA/SA mixtures (Figure 6), as well as for a reference sample of pure 12-HSA molecules at *r* = 0.5. Indeed, the value of the bilayer thickness of 24 Å that we obtained through SANS is exactly similar to that obtained in References [3,39] in the case of pure 12-HSA at the *r* = 0.2, which we consider here, i.e., with an excess of counterions. Such a state does not correspond to the L_β_ gel case, where the bilayer had a much larger value of around 42 Å, exactly twice the fatty acid length, as obtained when ethanolamine and 12-HSA had equimolarity for *r* = 0.5. [38] Moreover, when counterions are in excess, it has also been shown through coupled DSC and structural measurements that the multilamellar tube/micelle transitions matched the fluid/gel transition of the chains in the case of 12-HAS [3,39]. Thus, we hypothesize that fatty chains are present in a crystalline gel state in pure 12-HSA system at every *r*, but with very different crystalline structures, regardless of whether ethanolamine and 12-HSA have equimolarity or an excess of ethanolamine, with a strong interdigitation of the fatty acid chains within the gelled bilayer in this latter case. All spectra show some Bragg diffraction peaks at the large q, demonstrating that the fatty acids are in a crystalline gel state in all cases; the main correlation peak of the lamellar phase and its harmonics, as already evidenced using SANS, are also visible at low q. For the pure suspensions of 12-HSA molecules, the crystalline structure of the bilayer at *r* = 0.2 (R = 0) is strikingly different from the likely L_β_ gel phase reference at *r* = 0.5, confirming our hypothesis that the 12-HSA/ethanolamine ratio tunes the crystalline structure. The *r* = 0.2 solution indeed displays an intense peak at 1.495 Å^−1^ and a less marked one at 1.58 Å^−1^ within the q-window we examined, while the r = 0.5 shows an intense peak at 1.391 Å^−1^ and two others of lower intensity at 1.572 Å^−1^ and 1.597 Å^−1^. The pure SA system (R = 0) has another crystalline structure with a single rather broad Bragg peak at 1.53 Å^−1^.

The mixtures show four distinct behaviors. With the large excess of 12-HSA molecules (R ≥ 0.75), the suspension displays the same Bragg peaks as seen in in the R = 1 case. Hence, the introduction of a limited content of doping SA molecules does not prevent the system from crystallizing within the structure of the recovered pure 12-HSA solution. The amplitude of the peaks, however, decreases with an increase in SA content, suggesting that the size of the crystalline domains is reduced. Symmetrically, with a large excess of SA molecules (R = 0.05), the influence of the introduction of a small amount of 12-HSA doping molecules is minute since the structure is similar to that of pure 12-HSA. With an excess of SA fatty acids (0.1 ≤ R ≤ 0.5), the main crystalline structure recalls that of the pure SA sample: the Bragg peak of the SA remains and those of the 12-HSA crystalline structure at *r* = 0.2 are lacking. The intensity of such an SA Bragg peak decreases with an increase in R, showing the progressive decrease in the size of the crystalline phase in the lamellar domains of SA molecules. At the same time, a peak of low intensity appears at ~1.39 Å^−1^ whose amplitude increases concomitantly with the decrease in the main broad peak of SA with increasing R. At R = 0.5, both peaks have almost the same intensity. The origin of this new Bragg peak is likely the formation of crystal domains that contain the 12-HSA molecules and coexist with domains of pure SA molecules. Such a local phase separation between small domains that would have variable 12-HSA/SA ratios is consistent with the confocal microscopy and cryo-TEM experiments. The structure of the new crystalline phase that contains the 12-HSA molecules is difficult to assess, but it likely contains some SA molecules embedded in the 12-HSA molecules since some of the Bragg peaks obtained for the pure solutions of 12-HSA are not present on the diffractogram, whether for *r* = 0.2 or *r* = 0.5. Finally, at R = 0.6, the spectra can be viewed as a linear combination of the spectra of both pure systems of 12-HSA at *r* = 0.2 and SA, because it shows nicely all the Bragg peaks associated with their respective crystalline structures, while the Bragg peak of low intensity arising at lower R at 1.39 Å^−1^ is not any longer present. Moreover, the intensities of the main peaks of both structures are similar. This suggests the coexistence of similarly sized local domains of either pure 12-HSA or pure SA.

In summary, the crystalline structure of the gelled bilayers shifts continuously from that of the pure SA molecules to that of the pure 12-HSA molecules with the progressive introduction of 12-HSA molecules and a partial demixion between two types of molecule at intermediate R.

### 2.6. Rheology

Rheology was used to correlate the variety of structures observed as a function of R in terms of their mechanical properties. Shear–stress–amplitude sweep measurements were carried out at a frequency of 1 Hz for each ratio R from 0 to 1. All results are reported in Appendix A. Those measurements allow for the determination of the storage G′ and the loss G″ modulus of the samples, which, respectively, represent the solid/elastic and viscous components of the samples’ mechanical responses. Hereafter, we focus on the properties in the viscoelastic linear domain (Figure 7A,B and Appendix A).

Figure 7A displays the evolution of the elastic modulus G_0_′and viscous modulus G_0_″ determined for the linear domain as a function of R. The loss factor tan δ is shown as an inset for the sake of clarity. Three domains of R with distinctive behaviors can be distinguished and correlate well with the main structures of the mixture self-assemblies.

For R = 1, the elastic modulus G′_0_ of the pure 12-HSA sample is significantly higher than the loss modulus and the loss factor tan δ = G″/G′ is low (<<1). Furthermore, two asymptotic behaviors can be identified for the stress τ as function of strain γ. At low γ, τ varies linearly with γ, and then as γ^α^, with α < 1, once a critical stress τ_y_ is reached. τ_y_ is determined from the crossing point of the two asymptotic curves, as exemplified in Figure 7B for R = 0.6. These results indicate that, at 1 Hz, a pure 12-HSA sample at 2 wt% is an elastic gel at low stress and begins to flow under large strain, as can be observed in Appendix A. Similar behavior was reported for the same system with multilamellar tubes but with a different ethanolamine/12-HSA ratio of *r* = 0.5 [47]. A solution of tubes can develop a yield stress at sufficiently high concentrations [47], which is different from the jamming of spherical objects, which occurs during random close packing.

For R ≥ 0.75, the introduction of a few amount SA molecules in tubes causes a decrease in G′_0_ compared to the R = 1 samples while G″_0_ stays unchanged_._ Thus, the loss factor increases accordingly but always remains strictly below 1, which indicates a gel-like behavior. As for R = 1, the samples display the features of yield stress gels, with a linear variation of τ(γ) at low γ, but the yield stress τ_y_ decreases. This evolution is consistent with the shortening of the multilamellar tubes when decreasing R that is identified through confocal microscopy. Given that the number of fatty acids in a solution is kept constant, the decrease in the length of multilamellar tubes in turn decreases the number of tube entanglements and, subsequently, G′_0_.

From 0.6 ≥ R ≥ 0.4, G′_0_ and G″_0_ are almost constant, with G′_0_ being slightly larger than G″_0_. The samples are then also found in a gel state in this R system, which is confirmed by the loss factor value that remains below 1. Similarly to the system with a large R, samples display the characteristics of yield stress gels, with a linear variation of τ(γ) at low γ. A decrease in R leads to both an increase in tan δ, i.e., a less solid-like behavior, and an increase in τ_y_. This specific behavior arises from the fact that part of the rigid tubes turns into more flexible ribbons that bind the tubes to form longer objects, as observed via confocal microscopy. The self-assembled objects are then more flexible than at larger R ratios but their overall length creates more entanglements.

Finally, in the system with a low content of SA (from 0.25 ≥ R ≥ 0), the behavior is completely different. Samples are no longer gels but viscous fluids, since G″ is larger than G′, the loss factor consequently rises above 1 (inset in Figure 7A). However, they have G′_0_ values that are much larger than in the system with a larger R where tubes are formed, with values that can reach 20 times those obtained for R ≥ 0.4 in some ratios (R = 0.15, R = 0.2 and R = 0.25), the maximum being at R = 0.25. This may come from the increased rigidity of the bilayers in this system, as revealed via SANS. In this system, planar bilayers are formed, whether they are pure 2D lamella at R = 0 or faceted objects where the planar part is bound by curved parts at other R ratios. These objects can entangle and organize themselves at a mesoscopic scale in large domains that are able to move collectively (as observed through confocal microscopy) and slide one on the top of the other. Such a slippage would then explain the viscous fluid behavior of the solution.

For all samples, oscillatory measurements were completed with a frequency sweep between 0.05 and 5 Hz, with a constant strain of 0.1%. They all display a shear thinning behavior (see Appendix A). This is consistent with the yield stress fluid aspect in the case of gels. Regarding the fluid viscous samples made of planar lamella, this may result from an induced transition into multi lamellar vesicles under shear stress. The monitoring of the moduli did not enable us to determinate any relaxation time in the explored frequency range.

In summary, the mechanical properties of the samples display two main behaviors: at low content in SA, the samples are viscous fluids, with large G′_0_, and the samples turn into gels at higher R ratios of lower G′_0_, with a transition located between R = 0.25 and R = 0.4.

## 3. Discussion: Formation of the Different Aggregates Morphologies

The combination of the structural experiments carried out at 20 °C at all relevant scales of the system enabled us to build a diagram of the system as a function of R at this temperature. The main structures are illustrated in Figure 8. Please note that for the sake of simplicity, we have chosen to show only one type of structure per R (tube, helical ribbon, faceted object, planar lamella) but several structures coexist in the solution in the transition zones from one regime of structures to another. These transitions zones are likely fairly broad with a progressive evolution from one morphology to another within such transitions zones. Overall, all structures are based on 2D lamella stacked bilayers and display a very broad polymorphism, depending on the ratio between the two types of fatty acids. At low R, when doping the SA molecules by a small amount of 12-HSA molecules, the planar multilamellar self-assemblies that are formed in pure SA solutions turn into complex multilamellar faceted structures with planar domains bounded by spherical curves domains. At a large R, when doping the 12-HSA molecules by a small amount of SA molecules, the multilamellar long tubes that are formed in the pure 12-HSA systems are preserved but their tube lengths decrease upon the addition of SA. At intermediate R, tubes and helical ribbons coexist and eventually fuse into spectacular, long 1D multilamellar objects made up of tubes bonded by the helical ribbons. WAXS experiments revealed that two types of bilayers with different crystalline structures are present at low and intermediate R, which suggest a partial demixion between the two types of fatty acids for these ratios. There would then be a local distribution of variable 12-HSA ratios within the different parts of the structures. At low R, the 12-HSA molecules would accumulate preferentially within the curved parts of the self-assembled structures. At intermediate R, the SA would accumulate preferentially within the helical parts. These different morphologies of the self-assemblies (from 1D multilamellar tubes to large domains of fatty acid lamellas) are associated with different clear-cut regimes of bilayers rigidities and drive the macroscopic rheological properties.

A first hypothesis to explain the occurrence of the various morphological transitions would be a variation of the ionization state of the carboxylated heads with R that would change the packing parameter. The pH is, however, almost constant for all samples at 10.85 ± 0.15 (see Table 1): all carboxyl heads are in their COO^−^ form and are negatively charged and probably associated with the ethanolamine counterion, since the pH remains almost constant for one sample to another at 10.85 ± 0.15. The fact that 2D self-assemblies are formed in every case originates then from the fact that all bilayers are in a crystalline gelled state and not in a fluid state, as demonstrated by WAXS; otherwise, spherical micelles would have been formed, as these are usually obtained when long-chain fatty acids are fluid and negatively charged. Additionally, the fact that all heads are negatively charged implies that the electrostatic repulsions are similar throughout the whole range of R, regardless of whether they occur at the molecular level between heads or at the colloidal level between charged bilayers. Such electrostatic forces cannot therefore be invoked to explain the strong variations of both the interlamellar layer *d*_inter_ and the rigidity of the bilayer that occur when varying the R ratio (Figure 5B,C).

Another possibility that would explain the partial local demixion between the two different fatty acids is a possible mismatch between the respective thicknesses of bilayers 12-HSA or SA when interacting with ethanolamine ions at r = 0.2, in spite of their alkyl chains having the same lengths (21 Å). The fatty acids indeed crystallize in bilayers with different in-plane structures (Figure 6). The thickness of the lamella is, however, the only structural parameter that remains constant for all R ratios at ~24 Å (Figure 5D) which reveals a very strong interdigitation of the crystallized fatty acids within the bilayer. For the pure 12-HSA system, such an interdigitation comes from a mechanism related to the way the fatty acids interact with the ethanolamine counterions because it occurs only in a large excess of ethanolamine, *r* = 0.2, but not at equimolarity, *r* = 0.5. For this latter ratio, when the heads are not fully charged, hydrogen bonds exist between them, which leads to tighter packing than at lower *r* when bilayers are non-interdigitated. The fatty acids crystallize in this non-interdigitated bilayer almost in the same way as in the monolayers of 12-HSA at the air/water interface. The Bragg peaks we obtained at *r* = 0.5 (respectively, 1.391 Å^−1^, 1.572 Å^−1^ and 1.597 Å^−1^) indeed almost correspond to those measured by GIXD on the monolayers [24,25], whose position vary slightly with the surface pressure and, therefore, with the tilt angle between the monolayer and the interface. The interdigitation process induces a full reorganization of the molecules, since the interdigitated and non-interdigitated crystalline structures are completely different, although it remains impossible at this stage to determine whether ethanolamine ions penetrate the bilayer or not. Given that the crystalline bilayers of all samples at an ethanolamine/12-HSA *r* = 0.2 are interdigitated roughly with the same thickness, regardless of R, it is likely that ethanolamine counterions interact in the same way with the fatty acids, regardless of whether they are SA or 12-HSA molecules. The interaction must then take place between the heads of the fatty acids and ethanolamine.

The formation of the different morphologies with R and their associated transitions are then fully driven by the specific mechanisms induced by the presence of the OH hydroxyl function on the 12th carbon of the 12-HAS and the possibility of the formation of hydrogen bonds with neighboring OH groups whenever possible. It is enthalpically favorable for the 12-HSA molecules to form homogeneous domains in which they have the same curvature and can interact with one to each other through the H-bonding of OH groups in position 12. The behavior of the mixtures is then driven by the competition between the mixing entropy, which favors a homogeneous distribution of two types of fatty acids within the bilayers, and enthalpic effects that would lead to the segregation of the pure domains of SA molecules or 12-HSA molecules. Thus, depending on R, a local demixion can occur within a given type of self-assembled object, as is the case in catanionic systems [4,18,54]. This likely arises in the system of low R within the faceted objects. The demixion can even lead to the formation of different types of self-assembled structures that coexist in solution, as is the case in the transition zone from faceted objects to tubes/ribbons around R = 0.4. Samples in which two very different types of self-assembled structures coexist may undergo a gravity-induced phase separation due to differences of densities, which is partially arrested by the high viscosity of the samples. This may then create the large macroscopic heterogeneities observed at R = 0.4 (see the inset in Figure 2) and this makes analysis of SANS data difficult for this system.

The simplest case is the low doping regime of 12-HSA multilamellar tubes through SA molecules (R ≥ 0.75). The doping molecules only create defects within the bilayer but do not modify the main arrangement of the self-assemblies: the crystalline structure of the bilayer of a pure 12-HSA system is preserved, and the tubes’ external radius and interlamellar distances are of the same orders. Their introduction induces a progressive decrease in the rigidity of the bilayer, since η progressively but slightly increases with R. This also induces a progressive concurrent increase in the tube radius and a reduction in the length of the tubes. It is likely that part of the SA molecules mainly localizes at the edges of the 12-HSA bilayers which roll to form the tubes, thus releasing constraints and therefore decreasing the bilayer rigidity. The progressive capping of the bilayers at the tube ends may also decrease their length.

At low R, in the regime of the low doping of SA self-assemblies via a small amount of SA molecules (0.05 ≤ R ≤ 0.25), the multilamellar faceted structures with planar domains bounded by spherical curves domains suggest a local partitioning of the two fatty acids, which is confirmed by the presence of two type of crystalline structures via WAXS. The SA molecules localize mainly in the planar domains which have the same curvature as fore the pure solution, and 12-HSA molecules localize in the curved domains, which make up the edges of the faceted objects. Enthalpic effects then predominate in this regime and lead to the formation of these domains enriched by 12-HSA molecules, where they are able to form a rigid H-bond network and impose their curvatures. Such domains then act as rigid rafts within the bilayers and, therefore, end up limiting membrane fluctuations. This may explain the high membrane rigidity evidenced in this regime compared to all other R ratios. In this regime, the interlamellar distance *d*_inter_ decreases significantly and linearly from R = 0 to R = 0.25. This may be explained by the specific topology of the self-assembled structures. Since *d*_inter_ is much larger in the pure SA lamellar phase than in the tubes (575 Å versus 240 Å), the junction of the planar lamellas of SA and bilayers of smaller interlamellar distance force them to approach each other to fit the mismatch. The system then adopts an intermediate *d*_inter_, accounting for the respective contents of the planar and curved domain parts, which explains why *d*_inter_ decreases linearly with R. The increased rigidity of the membranes may also result from the fact that the planar membranes of SA molecules are forced to move closer together.

When R tends toward equimolarity at an intermediate R, WAXS demonstrates that demixion between the fatty acids occurs, as for the low R case. The main geometrical structures of the self-assemblies remain, however, close to those observed at a large R, as in the case of multilamellar tubes doped by 12-HSA tubes. These are long tubular objects with very close rigidities and interlamellar distances, such as those in the case of R = 0.75, while the crystalline structure stays close to that of pure SA. The large level of doping of the 12-HSA tubes by SA leads to the formation of helical ribbons, as already observed by Fameau et al. in Reference [11]; this results from their partial unrolling when the SA domains of planar curvature are inserted within the tubes. The demixion between acids also leads to the formation of several types of tubular objects with various local contents in 12-HSA/SA ratios, from rigid to helical ribbons, which eventually merge to form very long objects of various rigidities along the tubes. Additionally, close to the transition towards a low R regime at R = 0.4, it is likely that such tubular objects coexist with the faceted objects seen at low R ratios.

## 4. Materials and Methods

### 4.1. Materials and Sample Preparation

12-Hydroxystearic acid (12-HSA) was purchased from Xilong Chemical Co., Ltd., (Shantou, Guangdong Province, China); stearic acid (SA) was purchased from Sigma-Aldrich (St. Louis, MO, USA); and ethanolamine from Aldrich Chemistry (≥99.5%) (St. Louis, MO, USA). H_2_O was provided via a Millipore system, and D_2_O was purchased from Eurisotop (Saint-Aubin, France).

The aqueous stock-solutions of 12-HSA and SA at concentrations of 100 g/L were prepared as follows. 12-HSA (respectively SA) and ethanolamine were first weighed accurately in a sample tube into which ultrapure water was further added to reach the targeted molar concentration of fatty acid. The mass of ethanolamine was adjusted to set the molar ratio surfactant/counter ion *r* to 0.2, calculated by *r* = n_fatty_acid_/(n_fatty_acid_ + n_ethanolamine_). Then, the solutions were stirred and heated to 70 °C in an oven for 2 h, a temperature at which solutions are non-viscous, and were vortexed to ensure homogenization. They all appeared clear and homogeneous at 70 °C.

The mixture samples were then prepared at a fixed concentration of fatty acid molecules of 20 g/L by mixing stock-solutions and ultrapure water in appropriate dilutions to reach the targeted ratio of the two fatty acids, *R*, defined by: (1)$$R=\frac{n_{HSA}}{\left(n_{HSA}+n_{SA}\right)}$$
where *n*_HSA_ and *n*_SA_ are the molar concentrations of the fatty acids. The mixtures were then heated in an oven at 70 °C for two hours and vortexed for multiple seconds to ensure homogenization. This concentration was chosen in order to compare the results with those of the literature, where a large corpus of knowledge exists for the pure 12-has molecule system [11,38,39]. In the case of SANS experiments, all samples were prepared in D_2_O at the same volume fractions.

We examined the following R values in the paper: 1 (pure sample of 12-HSA fatty acids as reference), 0.9, 0.75, 0.6, 0.5, 0.4, 0.25, 0.1, 0.05 and 0 (reference sample of SA fatty acids).

Throughout the experiments, we systematically heated the samples at 70 °C for homogenization before any kind of measurement.

Table 1 shows the pH of each sample, which was measured with a Jenway 3510 pH Meter at room temperature.

### 4.2. Confocal Microscopy

A Zeiss LSM 700 confocal system mounted on a Zeiss microscope with an AxioCam MRm and a ×100 oil-immersion objective was used to examine the samples with Nile Red (Aldrich Chemistry) at a proportion of 1 molecule of Nile Red per 2 × 10^4^ molecules of fatty acid. Each sample was mixed with Nile Red dye at 70 °C and then measured at an ambient temperature. The dye was excited by a laser at 514 nm.

### 4.3. Cryogenic Transmission Electron Microscopy (Cryo-TEM)

The cryo-transmission electron microscopy (Cryo-TEM) was performed at the Institut de Minéralogie, de Physique des Matériaux et de Cosmochimie (IMPMC), Sorbonne Université. Samples were left in vials for two days at least to equilibrate to room temperature before being pipetted from vials. For each sample, a drop of the solution was deposited on a “quantifoil”^®^ carbon membrane grid (Quantifoil Micro Tools GmbH, Großlöbichau, Germany). The excess of liquid on the grid was absorbed with a filter paper and the grid was quench-frozen quickly in liquid ethane to form a thin vitreous ice film. Once placed in a Gatan 626 cryo-holder cooled with liquid nitrogen, the sample was transferred to the microscope and observed at a low temperature (−180 °C). Cryo-TEM images were recorded on an ultrascan 2k × 2k pixels CCD camera (Gatan, Pleasanton, CA, USA), using a LaB6 JEOL JEM2100 (JEOL, Tokyo, Japan) cryo microscope operating at 200 kV with a JEOL low-dose system (Minimum Dose System, MDS) to protect the thin ice film from any irradiation before imaging and reduce the irradiation during the image capture.

### 4.4. Small Angle Neutron Scattering (SANS)

Small angle neutron scattering experiments were performed at the Institut Laue-Langevin (Grenoble, France) on the D11 diffractometer (DOI: http://dx.doi.org/10.5291/ILL-DATA.9-11-2041). We used four configurations (6 Å at 1.7 m, 6 Å 5.5 m, 6 Å at 20.5 m and 13 Å at 38 m) to reach a very broad q-range, from 5.9 × 10^−4^ Å^−1^ to 0.53 Å^−1^. The samples were prepared in D_2_O in order to optimize neutron contrast with the hydrogenated molecules and to reduce incoherent scattering as much as possible. We confirmed beforehand that the use of deuterated water instead of hydrogenated water would not modify the macroscopic aspect of samples. Samples were held in flat quartz cells (Hellma) with a 2 mm optical path.

All samples were measured at 20 °C. The temperature was set using a circulating bath which thermalized the sample holder into which samples were placed. Transmissions, the scattering of empty cell, ^10^B_4_C ceramics (neutron absorber to value ambient background of experiment), the scattering of hydrogenated water and the differential cross-section scattering of water were measured independently. The subtraction of parasitic contributions and normalization through water to take into account the detectors’ heterogeneities were applied to raw data using GRASP software (version Grasp Lockdown V.9.22e) in order to obtain the corrected data in absolute units (cm^−1^) [55]. Contributions from the solvent and incoherent scattering were then subtracted.

The fitting software used was SasView 5.0.4 (http://www.sasview.org/, accessed on 3 August 2022). Fitting models are detailed in the Appendix A.

### 4.5. Wide Angle X-ray Scattering (WAXS)

WAXS measurements were carried out on a Xeuss 2.0 instrument from Xenocs, which uses a microfocused Cu Kα source with a wavelength of 1.54 Å and a PILATUS3 detector (Dectris, Switzerland). The experiments were performed at a sample-to-detector distance of 350 mm with a collimated beam size of 0.8 × 0.8 mm to achieve a q-range of 0.038–1.78 Å^−1^. The solutions were poured into 1.5 mm glass capillaries that were placed onto a home-made sample holder thermalized with a circulating water flow coupled with a Huber bath, allowing the control of the samples’ temperature at 20 °C. The measurements were performed for 35 min per sample to achieve a good results. The respective scattering from an empty beam, an empty capillary and a dark field were measured independently and subtracted from the sample scattering, taking into account their relative transmission, and normalized via incident beam intensity to obtain scattering in absolute units (cm^−1^). A reference water sample measurement was measured independently. The contribution from water to the sample scattering was then subtracted thanks to the reference water sample.

### 4.6. Rheology

Rheological measurements were carried out on an Anton Paar MCR 302 rheometer equipped with cone-plate geometry (50 mm diameter, 2° angle). The temperature during the experiment was set to 20 °C and controlled through the lower plate using a Peltier device. The setup was covered with a hood to prevent the sample from drying.

Each solution was heated in an oven at 60 °C for at least 45 min, then manipulated whilst still hot. Under these conditions, all solutions were homogeneous and fluid. Once the gap set, the measurements were taken after a resting time of 10 min for recovery and temperature homogenization.

Two types of oscillatory measurements were successively performed for each sample: an amplitude sweep measurement at a fixed frequency of 1Hz, a strain varying from 0.1 to 10% and a frequency sweep measurement at a fixed strain of 0.1% (which corresponds to the linear plateau) from 0.05 Hz to 5 Hz. The whole sequence took less than 40 min to complete, and no drying was observed at the end of the experiments. This protocol was repeated 3 times with renewed samples in order to estimate error bars.

## 5. Conclusions

We demonstrated the proof-of-concept of the simple strategy that we propose for tuning the properties of aqueous solutions of long-chain fatty acids based on a simple mixture of a saturated fatty acid and its hydroxylated counterpart, with the only variation being the ratio R between the two types of fatty acids and keeping all other physicochemical parameters, such as the ionization state or concentration, constant. The specific effects induced by the variable amounts of hydroxyl functions along the fatty chains, which are prone to making hydrogen bonds, allowed the creation of a wide range of nanostructures, from flat lamellar phases to entangled faceted objects and multilamellar rigid tubes to very long tubular objects with rigid and more flexible parts. The rheological properties of the solutions correlate very well with the different morphologies and also display a broad range of behaviors, from viscous fluids to gels with a yield stress. Additionally, in the regimes of large R ratios, we demonstrated that the dimensions of the multilamellar tubes (radius and length) of 12-HSA fatty acids, as widely described in the literature, can be finely tuned by the partial doping through SA molecules, which, in turn, tunes their rheological properties by playing on the density of entanglements.

The richness of both the morphologies of self-assemblies and rheological properties of these aqueous mixtures of bio-based renewable stearic fatty acids, as well as the melting of self-assemblies into micelles upon an increase in temperature, gives them a huge potential as building blocks for the design of green thermo-responsive foams or emulsions. We will examine the behavior of the aqueous mixtures of SA/12-HSA fatty acids under a wide range of temperatures, with a specific focus on the lamellar–micelle transition, in a forthcoming paper.

## Figures and Tables

**Figure 1 molecules-28-04336-f001:**
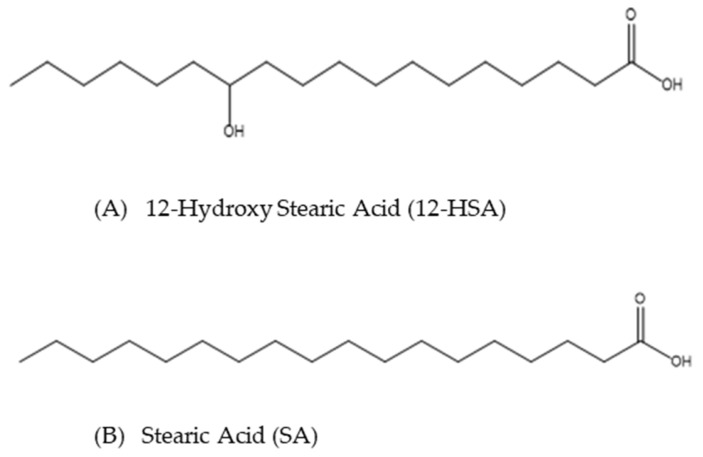
Representation of (**A**) 12-hydroxy stearic acid (12-HAS) molecule and (**B**) stearic acid (SA) molecules.

**Figure 2 molecules-28-04336-f002:**
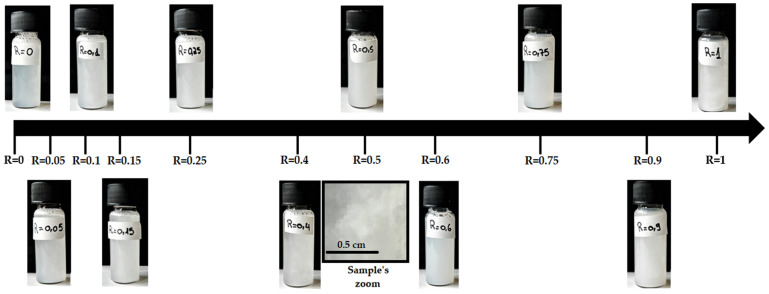
Photographs of the samples at all ratios R examined in the paper. The inset at R = 0.4 shows the heterogeneous macroscopic aspect.

**Figure 3 molecules-28-04336-f003:**
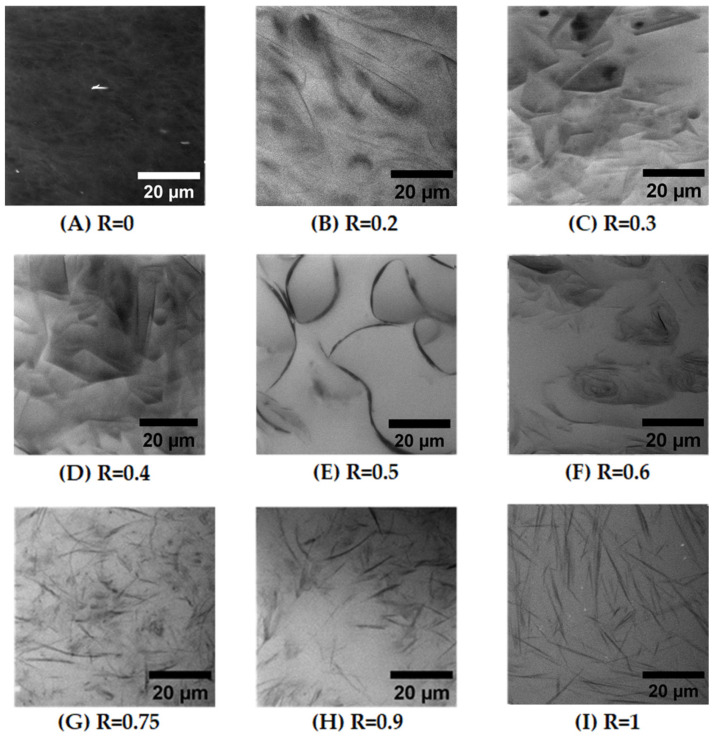
Confocal microscope images with an ×100 magnification obtained for different 12-has/SA samples at 2 wt% in the presence of Nile Red with different ratios to pure SA (R = 0) in figure (**A**) to pure 12-HSA (R = 1) in figure (**I**). Figures (**B**–**H**) correspond, respectively, to samples with R ratios equal to 0.2, 0.3, 0.4, 0.5, 0.6, 0.75 and 0.9.

**Figure 4 molecules-28-04336-f004:**
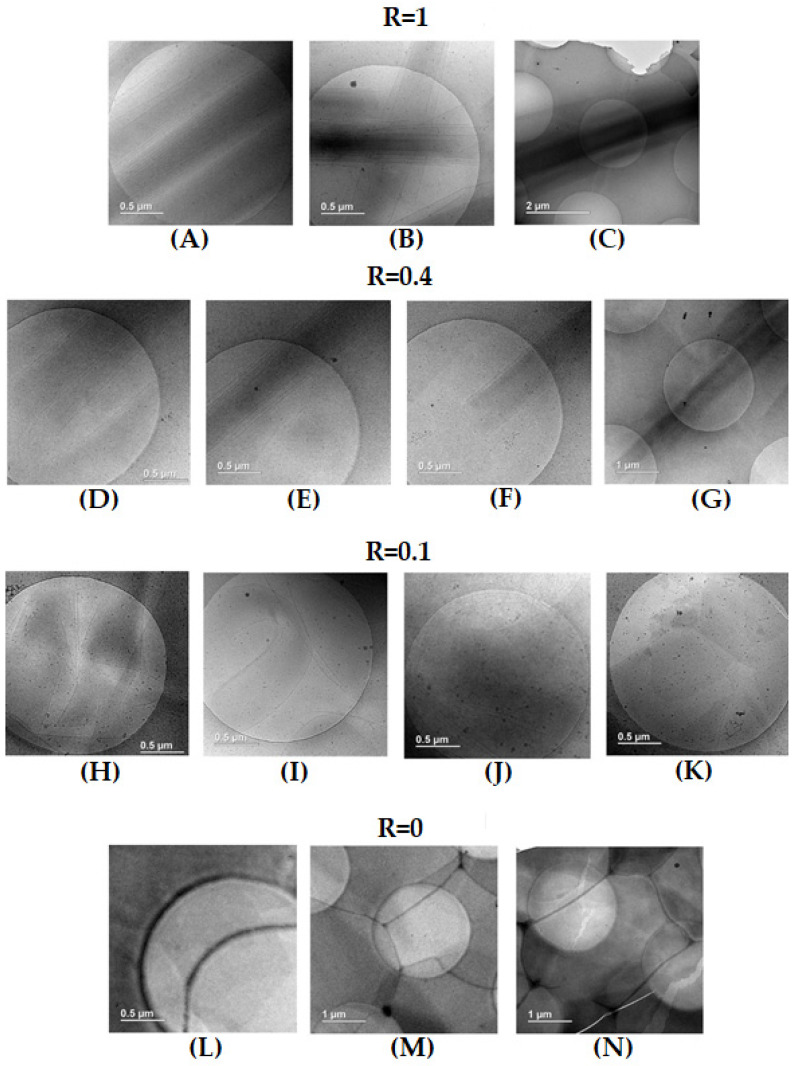
Cryo-TEM images obtained for different samples ratios from pure 12-HSA (R = 1, images **A**–**C**) to pure SA (R = 0, images **L**–**N**) and intermediate surfactant concentrations R = 0.4 (images **D**–**G**) and R = 0.1 (images **H**–**K**).

**Figure 5 molecules-28-04336-f005:**
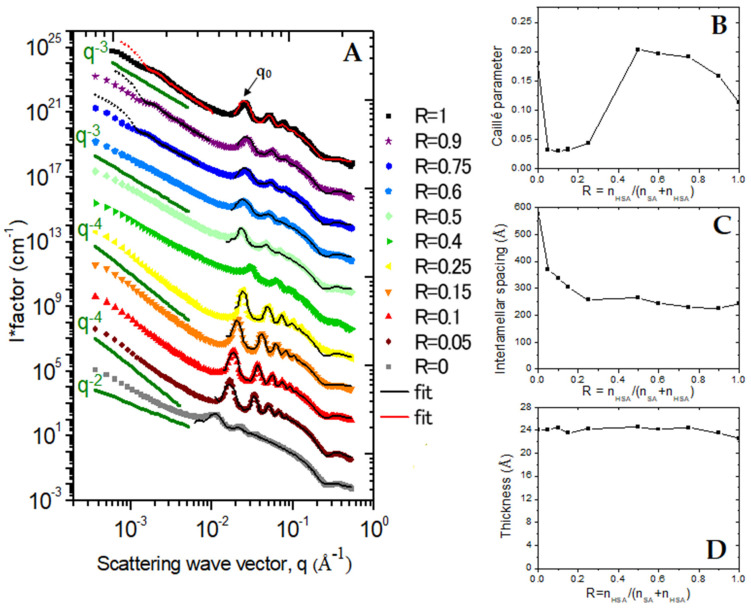
Right: (**A**) SANS intensity profiles at 20 °C for the different samples in D_2_O for all 12-HSA/SA ratios, from pure SA (R = 0) to pure 12-HSA (R = 1). The spectra are successively shifted by a factor of 10 in intensity for clarity (data for R = 0 on an absolute scale). The green lines correspond to the characteristic decays in the low q region. The black and red continuous lines correspond to the best fit of the data (see description of the model in Appendix A). The dotted part of the fitted model for R = 1, R = 0.9 and R = 0.75 correspond to the q-range where there is multiple scattering. The sample at R = 0.4 is not fitted as it is macroscopically heterogeneous (see main text). Left (**B**–**D**): Respectively, the Caillé parameters, interlamellar distances and lamella thicknesses as functions of R, obtained via SANS data fitting.

**Figure 6 molecules-28-04336-f006:**
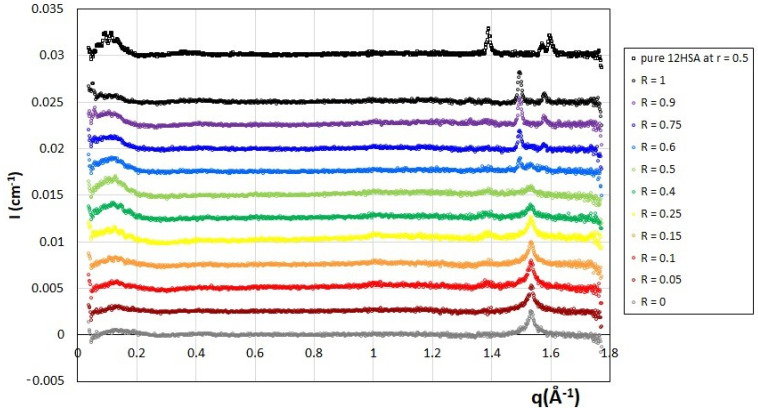
WAXS diffractograms for the different 12-HSA/SA sample ratios, from pure SA (R = 0) to pure 12-HSA (R = 1), as well as a pure 12-HSA sample with a fatty acid/ethanolamine ratio of r = 0.5 for comparison. For clarity, the spectra were shifted in intensity from each other by 0.0025 cm^−1^.

**Figure 7 molecules-28-04336-f007:**
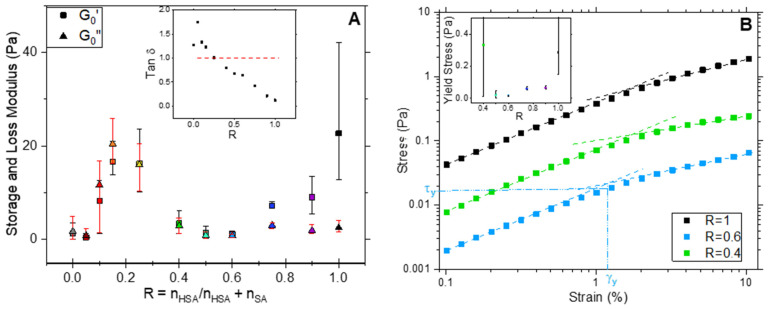
(**A**) Plateau value of storage G_0_′ and loss G_0_″ modulus measured at 1 Hz for the different sample ratios from pure SA (R = 0) to pure 12-HSA (R = 1) at 20 °C (loss factor in inset). (**B**) Shear stress as a function of the strain amplitude at a constant frequency of 1 Hz for selected samples (R = 0.4; 0.6 and 1). Asymptotic curves (dashed lines) depict the elastic regime followed by the solution yields. The associated yield stress is shown in the inset for R ranging from 0.4 to 1).

**Figure 8 molecules-28-04336-f008:**
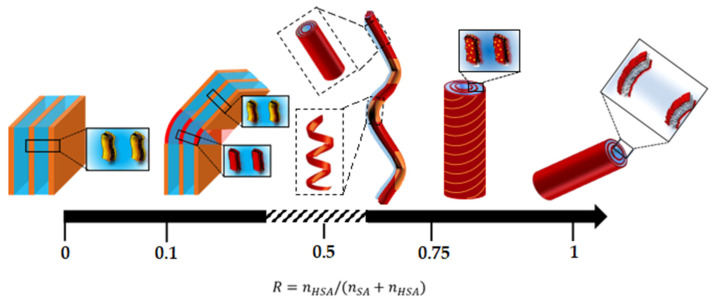
Schemes of the structures of the self-assembled aggregates as a function of R from pure SA (R = 0) to pure 12-HSA (R = 1) at 20 °C. The SA molecules are shown in orange and the 12-HSA molecules in red. The scale is not kept constant from one structure to another. The dashed part of diagram corresponds to the range of R where several structures coexist in the solution.

**Table 1 molecules-28-04336-t001:** pH values for each ratio (R), measured at room temperature.

R	pH
R = 0	[10.91;10.92]
R = 0.05	10.94
R = 0.1	[10.96;11.02]
R = 0.15	10.75
R = 0.25	10.93
R = 0.4	[10.75;10.79]
R = 0.5	[10.83;10.84]
R = 0.6	[10.86;10.88]
R = 0.75	[10.76;10.78]
R = 0.9	10.85
R = 1	[11.01;11.02]

## Data Availability

The raw data will be available from the corresponding authors upon reasonable request.

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
