# Peer review of "Aqueous Binary Mixtures of Stearic Acid and Its Hydroxylated Counterpart 12-Hydroxystearic Acid: Cascade of Morphological Transitions at Room Temperature"

_molecules, 2023, doi:10.3390/molecules28114336_

Round 1
Reviewer 1 Report
In this study, the authors present a straightforward strategy for tuning the properties of aqueous solutions containing long fatty chains by mixing a saturated fatty acid with its hydroxylated counterpart and adjusting the ratio (R) between the two types while keeping other physicochemical parameters constant. This approach takes advantage of the specific effects of hydroxyl functions, which can form hydrogen bonds, allowing for the creation of a diverse range of nanostructures, such as flat lamellar phases, entangled facetted objects, and multilamellar tubes with varying rigidity.
The rheological properties of these solutions correlate with their different morphologies, exhibiting behaviors ranging from viscous fluids to gels with a yield stress. In the regime of large R, the dimensions and rheological properties of 12-HSA fatty acid multilamellar tubes can be finely controlled by partially doping them with SA molecules. The authors used SANS to investigate the structure of the assemblies, WAXS to study the crystallinity of the fatty acid chains, and rheological experiments to determine the properties of the solutions.
One interesting finding was the increase in rigidity observed with small amounts of 12-HSA, as seen via SANS, which also correlated to an increased modulus. The paper is well-written, featuring extensive discussions and well-supported conclusions. Therefore, I recommend the publication of this work in Molecules.
One error that needs to be corrected is the absence of the hydroxyl moiety in figure 1 for the diagram of 12-HSA.
Reviewer 2 Report
In this work, the authors have investigated a simple strategy of tuning the morphological behavior of binary mixtures of fatty acid (stearic acid) and its hydroxylated counterpart in aqueous media at room temperature. The binary mixtures undergo a series of morphological transitions (from flat lamellar to entangled facetted objects and multilamellar rigid tubes to long tubular objects with twists) by varying the molar ratio of hydroxylated counterpart of stearic acid to stearic acid. The authors characterize these phases using sophisticated techniques to elucidate the mechanism of transition. The research is very interesting, fundamentally critical for advancing the understanding of “green” surfactant physical chemistry, and is of interest to the audience of MDPI molecules. However, there are few outstanding/minor questions that the authors are expected to address and highlight them in the manuscript before publishing.
After careful reading, I would like to offer my comments as follows.
1. The authors have provided an exhaustive introduction with relevant references. This is great to establish a background for the research work. However, in the current scenario, the novelty of the work is lost and the purpose of undertaking the present research is unclear. There seems to be a lack of logical flow.
To help make the distinction clear and bring out the work’s significance, it would be helpful to highlight the novelty using the proposed layout: (a) key highlights of the previous studies, (b) crucial points lacking in those studies, (c) importance of the missing pieces from a big-picture perspective, (d) how does the current study help to address those to advance the understanding, and (e) how does the current findings help the field – future scope and applications of the observations. This will help the reader to appreciate the current study and clarify the impact of the work.
2. In the Introduction section (line 144), please check if the citation is appropriate. I believe it should be Ref. 48 instead of Ref. 36.
3. In Figure 1, the OH functionality is missing for the chemical structure in (A). Additionally, the chemical names are incorrectly assigned to the structures in the figure caption. Please make the necessary changes.
4. In Section 2.4, it would be helpful to mark q0 in the Figure 5A plot. It would help the readers to follow along the conveniently.
5. In Section 2.6, please correct the figure label to Figure 7 (line 580).
6. Continuing on Section 2.6, can the authors provide some insights on why R=1.0 case shows higher elastic modulus from the perspective of the morphology of the self-assembled structure? Does the tubular morphology lead to greater probability of colloidal jamming to yield elastic gel-like nature compared to other morphologies?
7. Circling back to the original motivation presented by the authors about green surfactants and eco-friendly solutions, can the authors comment on how their findings aid towards the bigger picture? Can the authors provide some insight on what R ratios are desirable from an application standpoint and how readers can leverage this knowledge in designing surfactants? This is one of the most important points that I felt was not addressed by the authors while performing a deep dive into the fundamentals of the study.
Minor editing of English language required.
Reviewer 3 Report
In the manuscript "Aqueous binary mixtures of stearic acid and its hydroxylated 2 counterpart 12-hydroxystearic acid: cascade of morphological transitions at room temperature", the authors focus on synthesis of highly organized surfactant systems of various morphologies. These surfactant assemblies were synthesized by varying the ratio of natural occurring renewable components.
The authors use advanced experimental techniques to prove the structure of assemblies, which seem to be convincing and reveal a variety of morphologies that emerge at different concentration ratios of their building blocks.
Using renewable surfactants as an environmentally friendly alternative to synthetic amphiphiles is in the focus of modern green chemistry and materials science. The manuscript fits the scope of the special issue "Responsive Soft Materials Based on Biomolecules". It can be accepted for publication after the following issues are addressed:
1) The abstract seems to be too long and hard to read (380 words, while the template of "Molecules" recommends 200 words maximum). The authors should revise the abstract by shortening it and providing the relevance of the topic, the aim of the manuscript, a brief summary of its major results, and potential applications, if applicable.
2) Line 146. The authors state that they propose a new strategy to designing surfactant self-assemblies. Mixing reagents to create binary mixtures (as shown in Fig. 2) is, however, a traditional approach to synthesis of soft matter systems. Interactions between components mixed at various ratios have long been known to be fruitful for producing a variety of morphologies (for example, for polyelectrolyte-surfactant systems). It is recommended to clarify the novelty of the strategy the authors propose in this manuscript.
3) The differences between photos of samples in Fig. 2 are barely distinguishable with such a quality of images. All the mixtures are more or less turbid and homogeneous without substantial differences, so what was the purpose of demonstrating so many similarly looking photos? Is there a precipitate at R = 1? Do any other samples show a heterogeneous macroscopic structure represented by the R = 0.4 inset? In Line 199, heterogeneities were also reported for R = 0.5, but they are not distinguishable in Fig. 2. Please comment.
4) Line 182. The parameter "r" is misleading. Is it the same ratio "R" shown in Fig.2? Please clarify and correct, if applicable in the manuscript text (for example, in Lines 682 and 690).
5) Fig. 2 is given without any prior citing in the Results section. Fig. 3 is placed to the manuscript text before it is actually cited (Line 211). The authors should revise the manuscript accordingly to assure easier readability of all its figures by providing their correct citing in the text.
6) The formatting of the manuscript should be checked for its correspondence to the template of "Molecules" and applicable formatting styles, especially concerning citations in the manuscript text and numbering of references.
The quality of English is good. It is recommended to check the text for possible typos such as "OH hydroxyl function" (did the authors mean "OH hydroxyl group"?) in Lines 702-703.
